rsob.royalsocietypublishing.org

Subject Area:
molecular biology/microbiology

Keywords:
translation control, IRES elements, RNA structure

Author for correspondence:
Encarnacion Martinez-Salas
e-mail: emartinez@cbm.csic.es

# Deconstructing internal ribosome entry site elements: an update of structural motifs and functional divergences

Gloria Lozano, Rosario Francisco-Velilla and Encarnacion Martinez-Salas

Centro de Biología Molecular Severo Ochoa, Consejo Superior de Investigaciones Científicas—Universidad Autónoma de Madrid, Nicolás Cabrera 1, 28049 Madrid, Spain

iD EM-S, 0000-0002-8432-5587

Beyond the general cap-dependent translation initiation, eukaryotic organisms use alternative mechanisms to initiate protein synthesis. Internal ribosome entry site (IRES) elements are *cis*-acting RNA regions that promote internal initiation of translation using a cap-independent mechanism. However, their lack of primary sequence and secondary RNA structure conservation, as well as the diversity of host factor requirement to recruit the ribosomal subunits, suggest distinct types of IRES elements. In spite of this heterogeneity, conserved motifs preserve sequences impacting on RNA structure and RNA–protein interactions important for IRES-driven translation. This conservation brings the question of whether IRES elements could consist of basic building blocks, which upon evolutionary selection result in functional elements with different properties. Although RNA-binding proteins (RBPs) perform a crucial role in the assembly of ribonucleoprotein complexes, the versatility and plasticity of RNA molecules, together with their high flexibility and dynamism, determines formation of macromolecular complexes in response to different signals. These properties rely on the presence of short RNA motifs, which operate as modular entities, and suggest that decomposition of IRES elements in short modules could help to understand the different mechanisms driven by these regulatory elements. Here we will review evidence suggesting that model IRES elements consist of the combination of short modules, providing sites of interaction for ribosome subunits, eIFs and RBPs, with implications for definition of criteria to identify novel IRES-like elements genome wide.

## 1. RNA structural elements and gene expression regulation

In recent years, the impact of RNA structure in multiple steps affecting gene expression control has become increasingly evident. Key to understanding the relevance of RNA in biological processes has been the development of RNA structure-related methodologies, which has shown the flexible conformation of RNA, and also the modular nature of RNA three-dimensional (3D) architecture [1]. In particular, the advent of highly potent techniques, such as cryo-EM, provided critical insights into the 3D structure of reconstituted ribosome–RNA complexes [2,3]. In addition, implementation of novel techniques to analyse the RNA structure of ribonucleoprotein (RNP) macromolecules at the level of nucleotide resolution *in vitro* and also inside cells by selective 2′OH acylation analysed by primer extension (SHAPE) has revolutionized the field [4,5]. In this methodology, the availability of distinct probing reagents has been aided by the development of faster, reliable methods of analysis either via capillary electrophoresis or next-generation sequencing [6].

The molecular basis of RNA conformational flexibility has been extensively studied in viral RNAs. Generally, SHAPE models largely recapitulate RNA structures predicted by other methods, but also allow identifying unpredicted

rsob.royalsocietypublishing.org    Open Biol. **8**: 180155

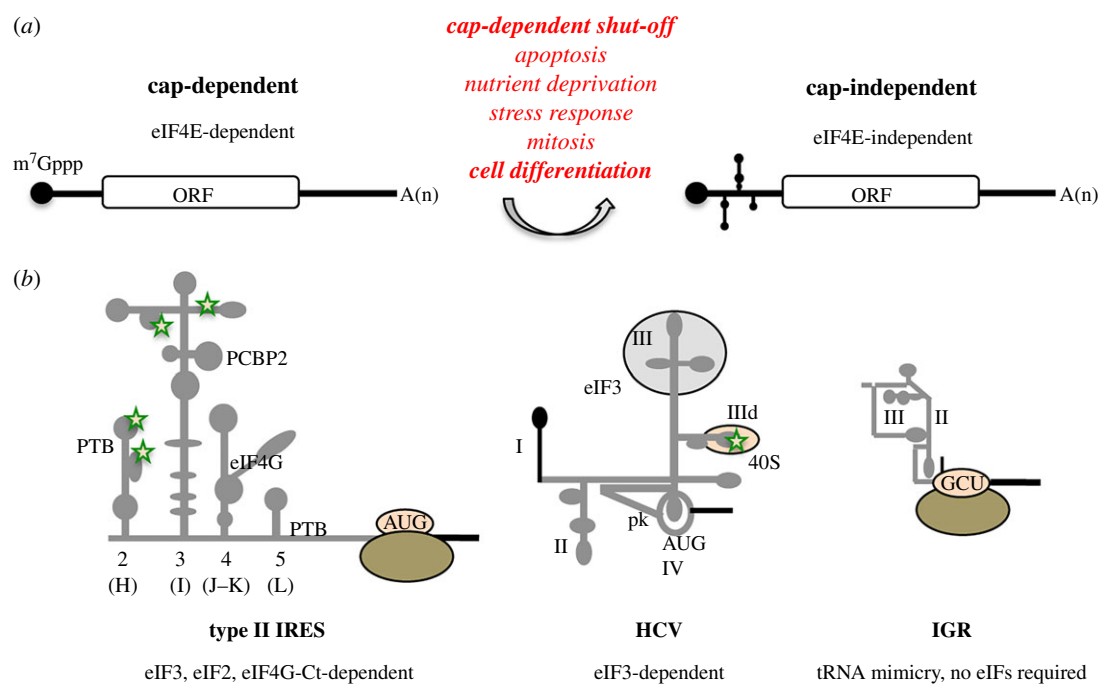

**Figure 1.** (*a*) Switch from cap-dependent to cap-independent translation. (*b*) Main features and secondary structure of the picornavirus type II IRES, the HCV IRES, and the dicistrovirus intergenic region (IGR) are represented. The location of domains referred to in the text is indicated. Green stars denote the location of sequences with modified conformational flexibility upon IRES incubation with ribosomal subunits. For type II IRES, the binding site of PTB, PCBP2 and eIF4G are indicated, while for HCV the binding site of eIF3 and the 40S recognition site are depicted.

structural elements, with novel regulatory properties [7–9]. This type of study is instrumental to explore new RNA motifs in RNA viruses, which despite apparent similar genomic architecture are divergent structurally. RNA structural elements, and their interactions with RNA-binding proteins (RBPs) and other ligands, control many stages of virus life cycle [10–13]. Along this line, structural studies combined with covariation analysis provided strong evidence suggesting selection pressures for functional elements in RNA viruses, designated internal ribosome entry site (IRES) elements, usually located in untranslated regions (UTR) of the mRNA [14,15]. However, evidence for structural elements within the open-reading frame (ORF) of viral RNAs has also been reported [16,17]. Conversely, structural divergence during evolution can generate new functional features, such as those shaping interactions with the host immune system or co-evolving with viral proteins [18]. Thus, although several RNA elements critical for the viral life cycle have been experimentally determined, the structure–function relationship of RNA motifs still remains to be understood.

Here we will focus on RNA motifs conserved in model IRES elements and how this information can be crucial to understand the modular organization of these RNAs, eventually contributing to the accurate prediction of IRES-like motifs in genomes. More detailed information on non-canonical and cap-independent translation mechanisms can be found in previous reviews [19,20].

## 2. Translation control. Evidence for cap-independent translation mechanisms

In all organisms, translation control is a key step in gene expression regulation. Eukaryotic mRNAs usually contain the $m^7G(5')ppp(5')N$ modification (or cap) at the 5' end (figure 1*a*). This structure mediates initiation of translation via the so-called cap-dependent mechanism that involves the binding of the translation initiation factor (eIF)-4E in a complex with eIF4G and eIF4A to the 5' end of mRNAs [21]. This complex recruits the 40S ribosomal subunit, in turn bound to the multimeric factor eIF3, eIF2 and the initiator met-tRNA$_i$, scanning the 5' UTR until an AUG triplet is found in the appropriate context to start protein synthesis. Joining of the 60S ribosomal subunit follows this step, producing a translation competent complex. It is well documented that strong cellular stresses (such as apoptosis, nutrient deprivation or oxidative stress) severely compromise cap-dependent translation [22]. However, under adverse situations specific subsets of cellular mRNAs remain associated to polysomes achieving efficient translation [23,24].

Viral mRNAs have developed distinct strategies to overcome the shut-off of cap-dependent protein synthesis induced in virus-infected cells [25]. Hence, although the cap-dependent mechanism was considered the predominant manner to initiate translation, several alternatives have been documented explaining selective translation of specific subsets of mRNAs [26]. In addition to RNA modification by methylation [27], selective translation of mRNAs enabling initiation at downstream codons encoding small ORFs in some cases bypassing upstream open-reading frames (uORFs), as in circular mRNAs and long non-coding RNAs, would rely on cap-independent mechanisms [28,29]. Moreover, while the monocistronic nature of eukaryotic mRNAs was historically considered the main source of protein coding genes, an increasing number of reports provided evidence for the expression of more than one ORF from a single transcriptional unit [30]. The presence of polycistronic RNAs is particularly evident in *Drosophila melanogaster* [31,32]. Therefore, as the annotation of genes in higher eukaryotes gains completeness and accuracy, initiation of protein synthesis using cap-independent mechanisms is becoming more frequent than initially thought.

Alternative manners to initiate translation received support from experimental evidence for distinct ribosome composition

rsob.royalsocietypublishing.org   Open Biol. 8: 180155

within the large population of ribosomal particles [33,34], also consistent with changes in the protein pattern detected by bi-dimensional gel electrophoresis in yeast with altered levels of the stalk proteins [35]. Specifically, analysis of the composition of ribosomal particles demonstrated the involvement of RACK1-containing ribosomes in the stimulation of translation promoted by two different viral IRES, hepatitis C virus (HCV) and cricket paralysis virus (CrPV) [36]. In agreement with the role of RACK1 in selective translation, this protein interacts with eIF3j, a peripheral subunit of eIF3 present in sub-stoichiometric quantities and subjected to post-translational modifications. More recently, implementation of improved hybrid mass spectrometry methods, in conjunction with advanced high-resolution cryo-EM, demonstrated that chloroplastic 70S and human 40S and 60S ribosomal particles are heterogeneous both in protein association and post-translational modification [37]. The heterogeneity of the human ribosomal particles influences their differential association to prototype viral IRES elements. Indeed, experiments conducted with conditional knock-down ribosomal stalk proteins, P1 and P2, revealed enhanced activity of the picornavirus foot-and-mouth disease virus (FMDV) IRES, but had no significant effect on the HCV IRES function [38]. Hence, it may be envisioned that the heterogeneous composition of ribosomes would also affect cellular IRES activity. Furthermore, not only modification of the ribosomal proteins but also modification of the ribosomal RNA impacts on the translation capacity of ribosomes [39]. Conversely, post-transcriptional modification of the mRNA has contributed to propose the need for distinct translation initiation mechanisms as well [27,40]. In these cases, the use of mRNA harbouring IRES elements has been instrumental to illustrate the differences between conventional and alternative mechanisms of protein synthesis initiation.

## 3. Viral IRES elements: impact of RNA structure on IRES activity

The pioneering work performed 30 years ago with picornavirus RNAs, which are naturally uncapped, provided the foundations for mRNA regions termed IRES elements (figure 1b). Picornavirus RNAs contain long, highly structured 5′ UTRs able to recruit the 40S subunit using a cap-independent mechanism [41,42]. This property was later extended to other RNA viruses [43]. Cumulative data obtained over the last decade by different laboratories provided evidence for the relationship between RNA structure and function of viral IRES elements [15,44]. Notwithstanding, IRES elements lack conserved primary sequence, secondary RNA structure, and host factor requirement to recruit the ribosomal subunits. This heterogeneity opens the question of how such a variety of diverse RNA regions perform the same function. This issue has been studied at the molecular level in prototype viral IRES elements. Currently, there is experimental evidence for different mechanisms to initiate translation internally. The simplest one operates in the dicistrovirus intergenic region (IGR) and involves direct interaction of the IRES with the 40S subunit. A more complex mechanism relies on the recognition of the IRES by translation initiation factors, which mediate the recruitment of the 40S ribosomal subunit. Representative members of the latter group are the HCV IRES and the diverse IRES elements present in the RNA genome of picornaviruses. Yet there are important differences among them, which will be discussed below.

The secondary structure and protein interactions of picornavirus IRES elements have been extensively analysed although high-resolution 3D structures are still lacking. Because of their heterogeneity, picornavirus IRES elements are classified into different types, such that each type harbours a common RNA structure core maintained by evolutionary conserved covariant substitutions. Due to their high efficiency and complex requirement of factors, type I and II IRES are prototypes to study internal initiation mechanisms. Type I IRES occurs in enterovirus (poliovirus, PV), and type II IRES in cardiovirus (encephalomyocarditis virus, EMCV) and aphthovirus (FMDV) (figure 1b). Both types I and II are independent of eIF4E but require the C-terminal region of eIF4G, eIF4A, eIF2 and eIF3 to assemble 48S initiation complexes [45]. Specifically, type II IRES elements are arranged in modular domains designated 2 to 5, or H to L, in FMDV and EMCV, respectively [46]. Domain 2 contains a conserved pyrimidine tract at the tip of a hairpin that provides a binding site for the polypyrimidine-binding protein (PTB) protein [47,48]. Domain 3 is a self-folding cruciform structure; the apical region of this domain harbours a conserved and essential GNRA tetraloop that mediates tertiary interactions [49,50]. Domain 4 is arranged in two hairpin loops held by a conserved stem and an A-rich bulge, which contain the binding site for eIF4G [51,52]. Domain 5 consists of a conserved hairpin, followed by a conserved pyrimidine tract and a variable single-stranded stretch of nucleotides at its 3′ end [53]. This domain provides the binding site for eIF4B and PTB [54], besides other RBPs such as Gemin5 and G3BP1 [55,56].

In further support of the modular organization of picornavirus IRES elements, structural analysis of the region of type II IRES interacting with eIF4G revealed that the conserved A-pentaloop serves as a docking site for base-pair receptors that requires the concerted action of all subdomains, since subtle changes in the orientation abrogate the interaction with eIF4G [57]. Interestingly, the configuration of the A-pentaloop resembles the GNRA tetraloop [58], where the G is substituted by A-A dinucleotide. The similarity of the RNA structure of these essential motifs, present in two different RNA domains [52], raises the possibility that they could be derived from RNA modules subjected to evolutionary changes to acquire novel functions.

The HCV 5′ UTR (figure 1b) contains conserved sequences and stem-loops controlling viral RNA translation, replication and stability [43,59–61]. Specifically, the HCV IRES consists of three domains (II, III and IV) [62]. Domain II is involved in eIF5-induced GTP hydrolysis of eIF2, while domain III binds eIF3 and the 40S subunit using two distinct subdomains, IIIabc and IIId, respectively [63]. A distinctive feature of the HCV IRES is the lack of eIF4G need for function [64]. In addition, the HCV IRES 3D structure includes a pseudoknot, besides a series of stem-loops connected by 3- and 4-way junctions [3]. Remarkably, this IRES element adopts a flexible RNA structure in solution, composed of an ensemble of conformers made of rigid parts that can move relative to each other [65].

The structure–function relationship of the HCV IRES has been studied in great detail using transcripts corresponding to the IRES region alone, in RNA replicons or inserted into artificial constructs within reporter genes [66]. However, in the context of the viral genome, HCV IRES activity is influenced by far downstream sequences involving long-distant

rsob.royalsocietypublishing.org Open Biol. 8: 180155

interactions [67]. Concerning upstream sequences, recent data showed that the liver-specific microRNA-122 (miR-122), which is complementary to two adjacent sequences in the spacer between domains I and II [68], assists the folding of the IRES by suppressing energetically favourable alternative secondary structures involving the miR-122 binding region adjacent to the IRES region [69]. Interestingly, coevolution between miR122 and HCV-related viruses affecting cattle with liver tropism and miR122 binding sites have been reported [70], suggesting that this could be a widespread feature of the hepacivirus group.

Various RNA viruses have been described to contain HCV-like IRES elements that, however, display subtle differences in their RNA conformation, as illustrated by pestivirus, hepacivirus, and type IV picornavirus IRES elements [71–74]. Moreover, the advent of massive sequencing methodologies, and therefore the discovery of novel RNA viruses worldwide, greatly increased the so-called HCV-like IRES. For instance, the IRES reported in the 5′ UTR of the Equine non-primate hepacivirus (EHcV) RNA consists of three domains which are homologous to domains I, II and III of HCV, albeit structural differences on domain III correlate with a lower IRES activity [75].

Contrary to hepacivirus and picornavirus, whose positive-strand RNA genome encodes a single long polyprotein, the genome of dicistroviruses is a natural dicistronic mRNA, in which translation of each ORF is governed by distinct IRES with different RNA structure organization and mode of action [76]. Activity of the 5′ IRES depends upon eIF3, resembling the HCV IRES. In contrast, the IGR adopts a 3D structure consisting of a triple-pseudoknot (PK I, II, and III) that functionally substitutes for the initiator met-tRNA$_i$ directing translation initiation at a non-AUG triplet [14] (figure 1b). The 3D structure of the IGR-ribosome shows that PKI resembles a tRNA/mRNA interaction in the decoding centre of the A site, mimicking a pre-translocation rather than the initiation state of the ribosome [2]. Pseudo-translocation of the IGR by elongation factor 2 (eEF2) in the absence of peptide bond formation brings the first codon of the mRNA into the A site to start translation. Remarkably, structural studies of the IGR–ribosome complex have shown the active role of the IRES in manipulating the ribosome. Indeed, the inchworm-like movement of the IGR suggests that this mRNA suffers cyclic conformational changes coupled with ribosomal inter-subunit rotation and 40S head swivel [77].

IRES activity was also suggested to mediate initiation of translation of some retroviral mRNAs such as HIV-1 gag mRNA [78,79]. Again, it is worth noting the diversity of sequences, secondary structures and mechanisms promoting translation initiation in retroviral RNAs [80,81]. Not surprisingly, this diversity challenges the criteria by which IRES elements are defined. However, during evolution distinct types of RNA elements have been selected in nature to promote initiation of protein synthesis not only in dicistronic but also in polycistronic RNA viruses [82]. In fact, the implementation of potent sequencing methodologies is allowing the identification of novel viruses infecting all type of organisms, thereby increasing the number of different gene expression systems. This is also the case for RNA viruses infecting plants, which promote initiation of translation by cap-independent mechanisms, in many cases depending upon sequences located on the 3′ end of the viral RNA [83].

# 4. Conformational flexibility of IRES domains: building blocks for ribosome interaction

Very soon after the discovery of IRES elements, functional analysis showed that RNA structure determines the function of the vast majority of viral IRES [14,15,62]. Indeed, IRES elements harbour distinct secondary structure motifs connected by junctions that play an essential role in RNA folding [3]. Both the sequence of motifs exposed on loops and the junctions are conserved in the IRES region of field isolates of highly variable RNA viruses [84–86], implying that the secondary structure is evolutionary constrained to deliver its function. Fully consistent with the biological relevance of the flexibility of IRES elements, perturbation of the local flexibility of specific IRES domains by RNA ligands inhibits RNA translation [87,88].

Although the presence of stable stem-loops in IRES elements has been determined by several RNA probing methodologies [89], the nucleotides involved in the dynamic folding and in tertiary interactions of IRES elements remain poorly known, presumably due to the inherent flexibility of the molecules and the lack of easy-to-use reliable methods to detect weak transient interactions. Notwithstanding, identification of RNA junctions is a key step in the structural characterization of flexible RNA molecules [3,50,90,91]. In this regard, the development of novel di-metallic chemical compounds, based on di-ruthenium, allowed the identification of four-way and three-way junctions within the FMDV (figure 2a) and HCV IRES conformation in solution [92], which were consistent with results derived from independent experimental approaches [65]. Therefore, di-metallic chemical reagents offer a new tool to determine regions controlling the folding of flexible RNA molecules.

The function of RNA molecules depends on their 3D structure and their ability to acquire distinct conformations [93]. Conversely, in response to specific signals conformational transitions could be spatially and temporally tuned, enabling the assembly of RNP complexes in a hierarchical ordered manner [94]. As such, the RNA reactivity towards slow- and fast-reacting SHAPE compounds can provide information on nucleotides that undergo local conformational changes on long timescales and those involved in tertiary interactions, respectively [95,96]. In this regard, differential SHAPE analysis on the free FMDV IRES showed that nucleotides reaching the final conformation on long time scales are placed on domains 4–5 upstream of the start codon, while nucleotides candidate to be involved in tertiary interactions are placed on the apical region of domain 3 [97]. Subsequent analysis of the IRES conformational flexibility conducted in the presence of various ribosomal fractions illustrated two key features of the IRES region: ribosomes free of factors (salt-washed) modified the conformational flexibility of domains 2 and 3 of the IRES element (figure 2b), while native ribosomes induced additional structural changes within domains 4 and 5 on long timescales (figure 2c). Furthermore, supplementing salt-washed ribosomes with soluble factors, including eIFs and RBPs, restored the RNA conformation of the IRES incubated with native ribosomes [97], reinforcing the role of host factors in mediating IRES function [98–100]. Therefore, individual structural modules of the IRES could perform a different role in the recruitment of ribosomes and host factors.

rsob.royalsocietypublishing.org    Open Biol. 8: 180155

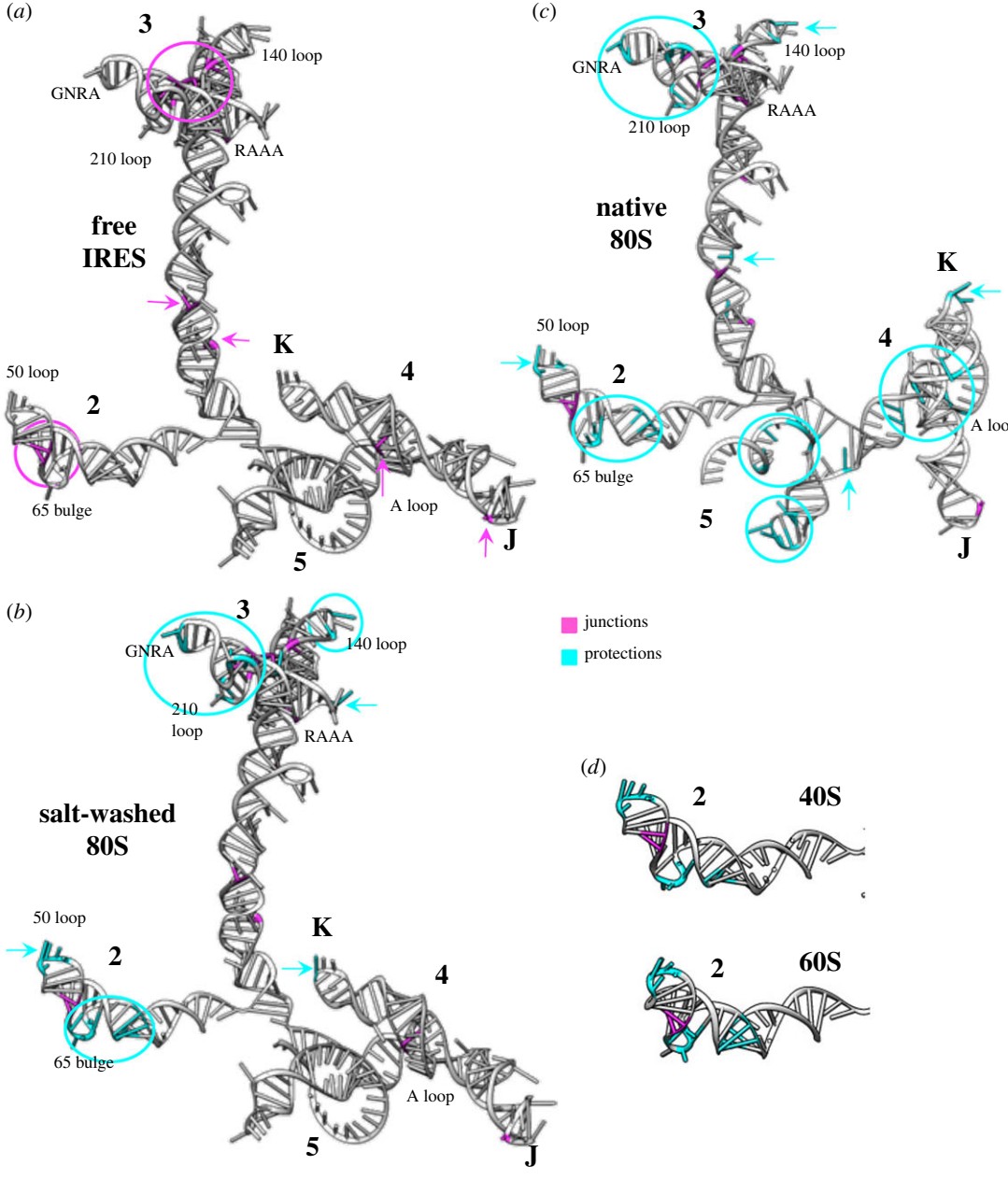

**Figure 2.** Conformational changes on the FMDV IRES induced by ribosomal fractions. Three-dimensional structure models for the IRES were predicted imposing SHAPE reactivity values obtained for the free RNA. The junctions defined by statistically significant reactivity towards di-ruthenium compound footprint are highlighted in pink (a), while the statistically significant reduction of SHAPE reactivity (protections) upon incubation with the ribosomal fractions is highlighted in cyan (b). Conformational changes observed by differential SHAPE, using isatoic anhydride (IA) treatment, upon incubation of the FMDV IRES with salt-washed 80S ribosomes (b), and native 80S (c). (d) Conformational changes observed in domain 2 upon incubation with purified 40S or 60S subunits. Domains 2, 3, 4 and 5, subdomains J and K of domain 4, as well as the GNRA tetraloop, loops and bulges referred to in the text, are indicated.

Experimental data accumulated over the years from many different laboratories have provided strong evidence for the involvement of RBPs on IRES function [101,102]. One of the putative roles played by RBPs on IRES activity is RNA chaperoning, stabilizing specific RNA conformations and thus allowing the interaction of eIFs with the IRES, as shown for PTB and Ebp1 on type II IRES elements [100]. In other cases, RBPs contribute to IRES activity removing RNA secondary structure near or at the start codon [103], or titrating IRES ligands as shown for Far upstream element-binding protein 1 (FBP1) [104], FBP2 [105] or SRp20 [106]. This is consistent with the fact that viruses inactivate the function of factors required for translation of the cellular mRNA competitors [107–109], as well as to inactivate negative regulators of IRES activity [110,111].

The conformational flexibility changes observed by differential SHAPE raise the possibility that the type II IRES contain separate sites for ribosome interaction and eIF binding. In support of this, direct interactions between 40S subunit and the EMCV IRES have been described [112]. These interactions are consistent with predicted base pairing between the HCV IRES and the 18S ribosomal RNA [113,114]. Interestingly, dissociated 40S and 60S ribosomal subunits, prepared from cells, induced fast structural changes within domain 2 (figure 2d) and the apical region of domain 3 of the FMDV IRES element *in vitro*. The presence of the FMDV IRES in mRNA expressed in living cells enhanced its association to the ribosomal subunits relative to a cap-mRNA. The enhanced IRES–ribosome association was observed both in normal conditions [97], and also upon stress induced by siRNA targeting eIF5B or poly-I:C

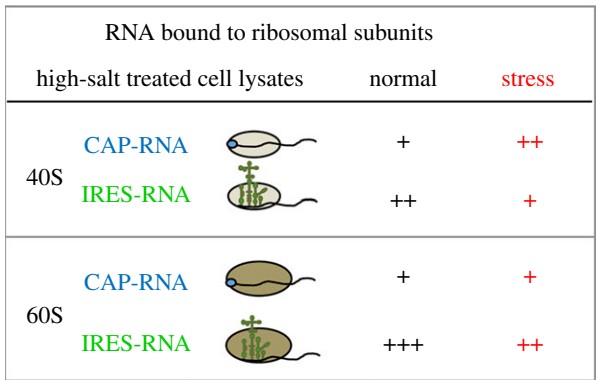

**Figure 3.** Ribosomal subunits association to cap-RNA or IRES-RNA expressed in human HEK293 cells. Stress cells refer to as siRNA targeting eIF5B or poly I:C treated cells.

treatment (figure 3), consistent with the capacity of the FMDV IRES to operate in normal cells but also upon strong stresses inhibitory for cap-dependent translation [25,101,102].

Therefore, taking into account that short conserved motifs within the type II IRES harbour the capacity to interact with the ribosomal particles, it could be concluded that RNA motifs present in domains 2 and 3 could define a functional building block responsible for the recruitment of ribosomal subunits. The remaining domains 4 and 5 would be mainly responsible for providing the binding sites for eIFs and RBPs, in agreement with previous reports [51,52,57,115]. Based on this data, we suggest that IRES elements could be derived from the association of distinct building blocks with specific features, such as those containing RNA motifs able to contact the ribosomal subunits. This module alone would not be sufficient to promote IRES activity, unless it is linked to other motifs facilitating the interaction with eIFs and RBPs. Individually, none of these building blocks contain full IRES activity, consistent with the observation that viral IRES elements function as single entities [116,117].

The finding that conserved RNA modules can provide direct IRES−ribosome interaction suggests that it could be possible to design synthetic RNAs with novel functional features, built from a combination of conserved building blocks connected via linker sequences to RNA motifs able to recruit transacting factors. These building blocks, however, should operate in concerted action to provide the correct and hierarchical orientation of the RNA motifs involved in ribosome and trans-acting factors recruitment. In support of the possibility to design synthetic RNAs behaving as IRES-like elements, it is worth mentioning synthetic RNA nanostructures recently designed by relying on the hierarchical formation of recurrent sequence-dependent networks of tertiary interactions [118]. These networks could specify RNA structural modules enabling orientation and topological control of helices to form larger self-folding domains.

Attempts to generate artificial IRES-like elements were previously reported [119,120]. In the first case, RNAs containing multiple copies of the motif carrying variations of the sequence (CCU)(n) function as artificial internal ribosome entry segments (AIRESs) in the presence of PTB. A different example of an artificial engineered RNA carrying five copies of a 9-nt motif (CCGGCGGGU) promoted cap-independent translation [120]. This motif, present in the cellular Gtx and RBM3 RNAs, was proposed to recruit ribosomes by binding directly to ribosomal proteins [121], using a

mechanism similar to hepacivirus and pestivirus IRES elements [64,122].

# 5. Cellular and viral IRES elements: similar RNA motifs with different regulatory functions?

Not surprisingly, the existence of cellular IRES elements is expected from the observation that novel mechanisms initially discovered in viruses have been invariably extended to the host cells. Hence, IRES activity has been claimed for a subset of cellular mRNAs [123,124]. Beyond atypical mRNAs characterized by having long and highly structured 5-UTRs, exhibiting translation under repressive conditions for general protein synthesis, the IRES elements of HOX cellular mRNAs were upregulated in different steps of embryonic development [23], therefore suggesting a role in normal gene expression programmes. This is also the case of the cofilin RNA, which is involved in the regulation of the axonal growth cone extension and turning [125]. Remarkably, the presence of distinct mechanisms to initiate translation could be instrumental to generate tools for therapeutic intervention, as illustrated by the treatment of spinocerebellar ataxia type 6, targeting the CACNA1A IRES element [126].

Yet the number of cellular IRES elements is rather reduced [127], especially relative to the potential ORFs in the genomes of high eukaryotic organisms. The lack of well-defined criteria to identify functional IRES elements in mRNAs promoting cap-independent translation could be due to several reasons. IRES-like elements may remain undetectable in genomes due to the lack of reliable tools and accurate methods to predict them [128,129], but also due to the difficulties to measure their activity, which might be detectable only under certain conditions. This was illustrated in a recent study focused to identify the RNA partners of eIF3, which allowed the identification of a selective group of cellular mRNAs translated in eIF3-depending manner [130]. In addition, identification of IRES elements in newly sequenced genomes of distinct organisms and their pathogens depend on the accurate annotation of coding genes, beyond the development of high-throughput methods to detect IRES activity. For instance, functional IRES elements were identified in viruses infecting filamentous fungi using a luciferase dual reporter system, including positive-sense RNA viruses belonging to the picornavirus-like group, non-segmented and tetra-segmented dsRNA viruses [131].

Worth mentioning are the efforts to implement specific methods for the computational search of sequences in mRNAs promoting cap-independent translation [132–134]. Early works to identify IRES elements widespread in cellular mRNAs took advantage of highly active viral proteases, which cleave eIF4G, and thus induce the shut-off of cap-dependent translation [135,136]. Likewise, the proteolytic activity of caspases in apoptotic cells provided a useful tool for the identification of mRNAs translated in a cap-independent manner [137]. In recent years researchers took advantage of mRNA display, a cell-free system for covalently linking newly translated proteins to their encoding RNA message [138]. Another study exploited the expression of bicistronic RNAs containing a combinatorial library of human sequences on the intercistronic space, followed by flow cytometry

rsob.royalsocietypublishing.org  Open Biol. **8**: 180155

separation of cells expressing tagged fluorescent proteins and deep sequencing of the mRNAs present in the fluorescent selected cells [139]. While these high-throughput works have tried to validate the accuracy of the method with a short list of the RNAs identified, there is still a long way to go to fix the criteria defining functional IRES elements.

A follow-up work aimed to develop methodologies for genome-wide computational prediction of IRES elements, which relied on the sequences shared by cap-independent translated mRNAs [139], resulted in the identification of very short motifs (C/U k-mers, 4 nt long) [134], thereby unlikely to be unique predictors for IRES elements given the size of mammalian mRNAs and the expected frequency of 4-nt motifs. These pyrimidine-rich motifs presumably provide binding sites for PTB and poly(rC)-binding protein 2 (PCBP2), frequently found interacting with viral IRES elements albeit with different functional relevance for IRES activity [98,99,115,140–142] (figure 1b). Moreover, a number of cellular 5′ UTRs that harbour $(CCU)_n$ sequences were reported to contain PTB-dependent IRES elements, raising the possibility that PTB or its interacting protein partners could provide a bridge between the IRES and the ribosome [119]. As such, it could be proposed that identification of RNA-binding sites for proteins involved in IRES function could be a clue to predict novel IRES-like motifs. Yet it is well established that various RBPs interact in a concerted manner with IRES elements, such that binding of one factor is not enough to promote IRES activity [143–145]. Hence, more work needs to be done to understand the function and structural organization of the diverse catalogue of IRES elements to better define the criteria in order to improve the accuracy for genome-wide prediction of IRES-like motifs.

## 6. Conclusion

Recent studies on IRES–ribosome complex formation have shown the active role of the IRES RNA in manipulating the ribosome. However, the ongoing IRES research is challenging mostly due to the structural diversity of the established viral IRES. In turn, the heterogeneity of IRES elements points to different strategies developed by viruses to exploit the host translation machinery. Data reported over the years have provided insights for different IRES-driven mechanisms to initiate translation. The simplest one involves direct recruitment of the ribosomal subunits mimicking features of the translation machinery, such as tRNAmet$_i$. However, the vast majority of IRES elements use more complex strategies involving the concerted action of eIFs and distinct RBPs to capture the 40S subunit, followed by the assembly of the 60S subunit. We would like to propose that complex IRES elements share conserved motifs which could behave as building blocks, enabling interaction with the ribosome. Eventually assisted by eIFs and RBPs, these motifs could contribute to assemble ribosomal subunits in a translationally active complex. Given that RNA is a versatile molecule in structure and function, the presence of distinct conserved motifs opens new avenues for designing engineered IRES elements with novel translation regulation features.

Data accessibility. This article has no additional data.

Competing interests. We declare we have no competing interests.

Funding. This work was supported by MINECO (BFU2017-84492-R) and an Institutional grant from Fundación Ramón Areces.

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
