## [Reviewer comments · Open Biology]

Review History

RSOB-18-0155.R0 (Original submission)

Review form: Reviewer 1

Recommendation

Accept with minor revision (please list in comments)

Are each of the following suitable for general readers?

- a) **Title**
Yes
- b) **Summary**
Yes
- c) **Introduction**
Yes

Is the length of the paper justified?

Yes

Should the paper be seen by a specialist statistical reviewer?

No

Is it clear how to make all supporting data available?

Not Applicable

Is the supplementary material necessary; and if so is it adequate and clear?

Not Applicable

Do you have any ethical concerns with this paper?

No

Comments to the Author

Review RSOB-18-0155

Gloria Lozano, Rosario Francisco-Velilla and Encarna Martinez-Salas:

"Deconstructing IRES elements: an update of structural motifs and functional divergences"

In this review, the authors provide a very interesting and comprehensive overview over viral and cellular internal ribosome entry site (IRES) elements, with a focus on the possible modular organization of IRES elements by using "building blocks" of small RNA segments that act with primary and/or secondary structure elements. This is a very interesting view and, in my opinion, warrants publication in Open Biology.

The text develops several aspects of IRES elements over long passages, while, in relation that, the actual section on "building blocks" is a bit short. However, this largely reflects the current status of research.

I have only a few suggestions:

1. In Fig. 1B, the authors could take the chance to mark some of the building blocks they mention in the text.
2. In the Type II IRES in Fig. 1B, the domains may be labeled also with numbers (2, 3 etc., which actually are mentioned in the text).
3. I was surprised to see PCBP2 acting on Type II IRES domain 3. Perhaps I missed that, but the literature cited for that (Andreev, Kafasla, Sweeney) does not support that statement. Perhaps PCBP2 interaction refers to Type I IRES elements? The authors may either better support that statement or remove the PCBP2 label from Fig. 1B.
4. In Fig. 2, the highlighting colours may be somehow changed for enhanced visibility.
5. "Evidence" has no plural form. Please correct "evidences" (repeatedly in the text) to "evidence".
6. In line 143, "polypyrimidin-binding protein (PTB) may be corrected to "polypyrimidine-tract binding protein" (an additional "e" and a "tract").

7. The statement in line 157 that SL I of HCV is involved in replication is incomplete, also SL II is involved in replication regulation (overlapping with its function in translation regulation).

8. When it comes to controls for checking if an RNA segment indeed is an IRES element, the authors should not miss to cite the useful experimental suggestions by Richard Lloyd's group (van Eden et al., 2004) and perhaps also comment on the use of these criteria in recent high throughput experiments.

Decision letter (RSOB-18-0155.R0)

25-Sep-2018

Dear Dr Martinez-Salas

We are pleased to inform you that your manuscript RSOB-18-0155 entitled "Deconstructing IRES elements: an update of structural motifs and functional divergences" has been accepted by the Editor for publication in Open Biology. The reviewer(s) have recommended publication, but also suggest some minor revisions to your manuscript. Therefore, we invite you to respond to the reviewer's comments and revise your manuscript.

Please submit the revised version of your manuscript within 14 days. If you do not think you will be able to meet this date please let us know immediately and we can extend this deadline for you.

- 1) A text file of the manuscript (doc, txt, rtf or tex), including the references, tables (including captions) and figure captions. Please remove any tracked changes from the text before submission. PDF files are not an accepted format for the "Main Document".
- 2) A separate electronic file of each figure (tiff, EPS or print-quality PDF preferred). The format should be produced directly from original creation package, or original software format. Please note that PowerPoint files are not accepted.

3) Electronic supplementary material: this should be contained in a separate file from the main text and meet our ESM criteria (see <http://royalsocietypublishing.org/instructions-authors#question5>). All supplementary materials accompanying an accepted article will be treated as in their final form. They will be published alongside the paper on the journal website and posted on the online figshare repository. Files on figshare will be made available approximately one week before the accompanying article so that the supplementary material can be attributed a unique DOI.

Online supplementary material will also carry the title and description provided during submission, so please ensure these are accurate and informative. Note that the Royal Society will not edit or typeset supplementary material and it will be hosted as provided. Please ensure that the supplementary material includes the paper details (authors, title, journal name, article DOI). Your article DOI will be 10.1098/rsob.2016[last 4 digits of e.g. 10.1098/rsob.20160049].

4) A media summary: a short non-technical summary (up to 100 words) of the key findings/importance of your manuscript. Please try to write in simple English, avoid jargon, explain the importance of the topic, outline the main implications and describe why this topic is newsworthy.

Images

Data-Sharing

It is a condition of publication that data supporting your paper are made available. Data should be made available either in the electronic supplementary material or through an appropriate repository. Details of how to access data should be included in your paper. Please see <http://royalsocietypublishing.org/site/authors/policy.xhtml#question6> for more details.

Data accessibility section

Sincerely,

The Open Biology Team
<mailto:openbiology@royalsociety.org>

Reviewer's Comments to Author:

Referee: 1

Comments to the Author(s)
Review RSOB-18-0155

Gloria Lozano, Rosario Francisco-Velilla and Encarna Martinez-Salas:
"Deconstructing IRES elements: an update of structural motifs and functional divergences"

In this review, the authors provide a very interesting and comprehensive overview over viral and cellular internal ribosome entry site (IRES) elements, with a focus on the possible modular organization of IRES elements by using "building blocks" of small RNA segments that act with primary and/or secondary structure elements. This is a very interesting view and, in my opinion, warrants publication in Open Biology.

The text develops several aspects of IRES elements over long passages, while, in relation that, the actual section on "building blocks" is a bit short. However, this largely reflects the current status of research.

I have only a few suggestions:

1. In Fig. 1B, the authors could take the chance to mark some of the building blocks they mention in the text.
2. In the Type II IRES in Fig. 1B, the domains may be labeled also with numbers (2, 3 etc., which actually are mentioned in the text).
3. I was surprised to see PCBP2 acting on Type II IRES domain 3. Perhaps I missed that, but the literature cited for that (Andreev, Kafasla, Sweeney) does not support that statement. Perhaps PCBP2 interaction refers to Type I IRES elements? The authors may either better support that statement or remove the PCBP2 label from Fig. 1B.
4. In Fig. 2, the highlighting colours may be somehow changed for enhanced visibility.
5. "Evidence" has no plural form. Please correct "evidences" (repeatedly in the text) to "evidence".
6. In line 143, "polypyrimidin-binding protein (PTB) may be corrected to "polypyrimidine-tract binding protein" (an additional "e" and a "tract").
7. The statement in line 157 that SL I of HCV is involved in replication is incomplete, also SL II is involved in replication regulation (overlapping with its function in translation regulation).
8. When it comes to controls for checking if an RNA segment indeed is an IRES element, the authors should not miss to cite the useful experimental suggestions by Richard Lloyd's group (van Eden et al., 2004) and perhaps also comment on the use of these criteria in recent high throughput experiments.

Author's Response to Decision Letter for (RSOB-18-0155.R0)

See Appendix A.

RSOB-18-0155.R1

Review form: Reviewer 1

Recommendation

Accept as is

Are each of the following suitable for general readers?

- a) **Title**
Yes
- b) **Summary**
Yes
- c) **Introduction**
Yes

Is the length of the paper justified?

Yes

Should the paper be seen by a specialist statistical reviewer?

No

Is it clear how to make all supporting data available?

Yes

Is the supplementary material necessary; and if so is it adequate and clear?

Yes

Do you have any ethical concerns with this paper?

No

Comments to the Author

The authors have followed my recommendations, I am fine with the changes.

Decision letter (RSOB-18-0155.R1)

30-Oct-2018

Dear Dr Martinez-Salas

We are pleased to inform you that your manuscript RSOB-18-0155.R1 entitled "Deconstructing IRES elements: an update of structural motifs and functional divergences" has been accepted by the Editor for publication in Open Biology.

Sincerely,

The Open Biology Team
mailto: openbiology@royalsociety.org.

Decision letter (RSOB-18-0155.R2)

30-Oct-2018

Dear Dr Martinez-Salas

We are pleased to inform you that your manuscript entitled "Deconstructing IRES elements: an update of structural motifs and functional divergences" has been accepted by the Editor for publication in Open Biology.

Sincerely,

The Open Biology Team
mailto: openbiology@royalsociety.org

CENTRO DE BIOLOGIA MOLECULAR
"SEVERO OCHOA"

Appendix A

Royal Society's Open Biology
Editorial office

October 3th, 2018

Dear Dr. Glover,

We have submitted a revised version of the review entitled "Deconstructing IRES elements: an update of structural motifs and functional divergences" by Gloria Lozano, Rosario Francisco-Velilla, and myself, to be considered for publication in Open Biology.

We were very happy to see the positive comments of yourself and the reviewer. To address his/her points, we have modified the manuscript to include revised versions of Figures 1 and 2. A modification to the text was included in page 9 to better explain the section on RNA building blocks. All minor points were corrected. As a consequence of the changes inserted in the text several references were added.

A point-by-point response is shown in the next pages.

We thank you for your attention and look forward to hearing from you,

Encarna Martínez-Salas
Centro de Biología Molecular Severo Ochoa
email: emartinez@cbm.csic.es
<http://web4.cbm.uam.es/joomla-rl/index.php/en/scientific-departments/genome-dynamics-and-function?id=%20536>

E-mail: emartinez@cbm.csic.es

Centro de Biología Molecular "Severo Ochoa"
Consejo Superior de Investigaciones Científicas
Universidad Autónoma de Madrid
Cantoblanco, 28049. MADRID
TEL.: 91 1964619
FAX: 91 1964420

RESPONSE:

In this review, the authors provide a very interesting and comprehensive overview over viral and cellular internal ribosome entry site (IRES) elements, with a focus on the possible modular organization of IRES elements by using "building blocks" of small RNA segments that act with primary and/or secondary structure elements. This is a very interesting view and, in my opinion, warrants publication in Open Biology.

The text develops several aspects of IRES elements over long passages, while, in relation that, the actual section on "building blocks" is a bit short. However, this largely reflects the current status of research.

- Thanks for this comment. We have expanded this section including references to two examples of artificial IRES elements (see page 9)

I have only a few suggestions:

1. *In Fig. 1B, the authors could take the chance to mark some of the building blocks they mention in the text.*

-Done as requested.

2. *In the Type II IRES in Fig. 1B, the domains may be labeled also with numbers (2, 3 etc., which actually are mentioned in the text).*

-Done as requested.

3. *I was surprised to see PCBP2 acting on Type II IRES domain 3. Perhaps I missed that, but the literature cited for that (Andreev, Kafasla, Sweeney) does not support that statement. Perhaps PCBP2 interaction refers to Type I IRES elements? The authors may either better support that statement or remove the PCBP2 label from Fig. 1B.*

- Thanks for this comment. The figure was correct. However, as indicated in the text PCBP2 binds to both type I and type II although we failed to cite references for type II, which are now included.

4. *In Fig. 2, the highlighting colours may be somehow changed for enhanced visibility.*

-Done as requested.

5. *"Evidence" has no plural form. Please correct "evidences" (repeatedly in the text) to "evidence".*

-Done as requested.

6. *In line 143, "polypyrimidin-binding protein (PTB) may be corrected to "polypyrimidine-tract binding protein" (an additional "e" and a "tract").*

-Done as requested.

7. *The statement in line 157 that SL I of HCV is involved in replication is incomplete, also SL II is involved in replication regulation (overlapping with its function in translation regulation).*

- Thanks for this comment. We have modified the sentence, and the corresponding references are included.

CENTRO DE BIOLOGIA MOLECULAR
"SEVERO OCHOA"

8. *When it comes to controls for checking if an RNA segment indeed is an IRES element, the authors should not miss to cite the useful experimental suggestions by Richard Lloyd's group (van Eden et al., 2004) and perhaps also comment on the use of these criteria in recent high throughput experiments.*

- We fully agree with the reviewer. The reference by R. Lloyd team is now included.

E-mail: emartinez@cbm.csic.es

Centro de Biología Molecular "Severo Ochoa"
Consejo Superior de Investigaciones Científicas
Universidad Autónoma de Madrid
Cantoblanco, 28049. MADRID
TEL.: 91 1964619
FAX: 91 1964420